# A Power System Timing Data Recovery Method Based on Improved VMD and Attention Mechanism Bi-Directional CNN-GRU

Kangmin Xie, Jichun Liu * and Youbo Liu

School of Electrical Engineering, Sichuan University, Chengdu 610065, China
* Correspondence: jichunliu@scu.edu.cn

**Abstract:** The temporal data of the power system are expanding with the growth of the power system and the proliferation of automated equipment. However, data loss may arise during the acquisition, measurement, transmission, and storage of temporal data. To address the insufficiency of temporal data in the power system, this study proposes a sequence-to-sequence (Seq2Seq) architecture to restore power system temporal data. This architecture comprises a radial convolutional neural unit (CNN) network and a gated recurrent unit (GRU) network. Specifically, to account for the periodicity and volatility of temporal data, VMD is employed to decompose the time series data output into components of different frequencies. CNN is utilized to extract the spatial characteristics of temporal data. At the same time, Seq2Seq is employed to reconstruct each component based on introducing a feature timing and multi-model combination triple attention mechanism. The feature attention mechanism calculates the contribution rate of each feature quantity and independently mines the correlation between the time series data output and each feature value. The temporal attention mechanism autonomously extracts historical–critical moment information. A multi-model combination attention mechanism is introduced, and the missing data repair value is obtained after modeling the combination of data on both sides of the missing data. Recovery experiments are conducted based on actual data, and the method's effectiveness is verified by comparison with other methods.

**Keywords:** neural networks; VMD; data reconfiguration; attention mechanisms

## 1. Introduction

Power grid spatial and temporal character is becoming more complicated with the development of the power system, and the automation equipment rapidly expands with large-scale power systems [1,2]. At the same time, measuring data are increasing. They are starting to resemble big data due to the rapid advancement of power system measurement technologies and the ongoing reduction in measurement costs [3]. The transmission, storage, and analysis of massive amounts of big data for power grids have emerged as a significant area of research in recent years thanks to the rapid advancement of big data technology [4,5]. It is possible to estimate the status of the power system and equipment to a significant extent as well as to optimize operation and accident analysis through the analysis of massive and multiple types of time series data [6,7].

It goes without saying that obtaining authentic and accurate data is crucial for data processing. Still, since signal attenuation, interference, and occasionally failing electronic acquisition equipment cause data to be lost during data acquisition, measurement, transmission, and storage, it is impossible to obtain accurate time series data. In addition to complicating the analysis of prediction outcomes or trend development based on extensive data analysis, missing data can also have an impact on system state estimate, stability, and other critical features based on network data analysis [8,9]. The power system measurement configuration itself has a certain redundancy at the beginning of the design, and for

some of the missing time series data, under the premise of satisfying the observable state estimation, it can be replaced by pseudo-measurements or by data that are similar in time and space without causing an unacceptable impact on the overall system state accuracy. In addition, the state estimation through the system can be used as a basis for filling in the missing data [10]. However, when there is a large amount of missing time series data, state estimation by redundancy is not possible. Then, the missing data need to be repaired by mathematical or engineering means.

Numerous solutions to the missing data issue have been put forth by domestic and international researchers. The pre-processing category and the post-assessment category are the two main categories. On the basis of system timing data and system topology, the latter primarily builds the system state equation to recover the data. In the literature [11], a real-time dynamic parameter estimation method based on vector measurement unit (PMU) data with extended Kalman filtering (EKF) is proposed. In order to reduce the computational difficulty of this method, the literature [11] uses a model decoupling technique. However, this method is only applicable to the real-time estimation of the states and parameters related to electromechanical dynamics. The literature [12] proposes a robust detection method using temporal correlation and statistical consistency of time series data, offering three innovative matrices that capture measurement correlation and statistical consistency by processing predicted states and reliable information inserted from phasor measurement units. Pre-processing techniques are primarily employed to recover missing data from known data. The two main categories of preprocessing ideas are: (1) analyzing the characteristics of the data in the missing data domain to complete the data as described in the literature [13–16] and (2) analyzing the overall trend and overall structure of the data and completing the data [17–19]. Ref. [13] used a Lagrangian interpolation polynomial method for adaptive estimation of incomplete and missing data, but this method is limited to the case where there are few missing data.There are also some scholars who convert the measured value into a Page matrix and then use low-rank matrix estimation based on the optimal singular value threshold to reconstruct the original signal [14]. The literature [15] first proved that the power quality data have the property of approximately low rank. Based on this, a multi-parametric joint rank optimization model is designed, and the alternating direction multiplier method is applied to decompose it into several subproblems for solving separately. At the same time, the optimal selection strategy of adaptive iteration steps is proposed to speed up the model solution for the problem that the traditional alternating direction multiplier method solves slowly. Ref. [16] uses the singular value threshold algorithm to complete the missing data twice and analyzes its error on the basis of the completion. However, the above two methods are not effective for complex missing data. According to Ref. [17], a shallow coder is used to learn the data features, and after processing, the data are supplemented by weighting the data structure. According to Ref. [18], forward and reverse GRU networks are used separately to learn the existing data, and their combined results are then weighted to achieve the goal of completing the data. The above two methods have a large gap in the reconstruction effect of different types of data. Ref. [19] has constructed an improved generative countermeasure network learning time series data with complex time and space relations. According to the data's redundancy and inherent physical and mathematical relations, the data can be restored to a considerable extent. However, this method consumes many resources and could be more conducive to practical use. Table A1 in the Appendix A shows a summary of information for similar work.

Based on the above background and the time series and multidimensional correlation characteristics of power system timing data, a method for recovering missing data from power system measurements based on dual radial gated cyclic units is proposed. The method learns the spatiotemporal characteristics of the historical data, obtains sufficient generalization capability for the time series data, constructs a mapping of the existing data to the missing data, and makes this mapping select the relatively valuable information in the existing data to repair the missing data in real time through a triple attention mechanism.

In order to make full use of the existing information, this paper proposes a joint neural network approach, i.e., to build neural networks on both sides of the missing data separately and finally to obtain the weighted repair results by combining two neural networks. Finally, a comparison between simulated and actual data shows that this data-driven method of repairing missing data in power system measurements, which does not rely on the power system topology, can maintain a high accuracy rate under different numbers of absent conditions.

## 2. Power System Time Series Missing Data Recovery Model Structure

The method of time series data recovery of power systems based on VMD and triple attention mechanism bi-directional CNN-GRU is shown in Figure 1, which is divided into four main steps:

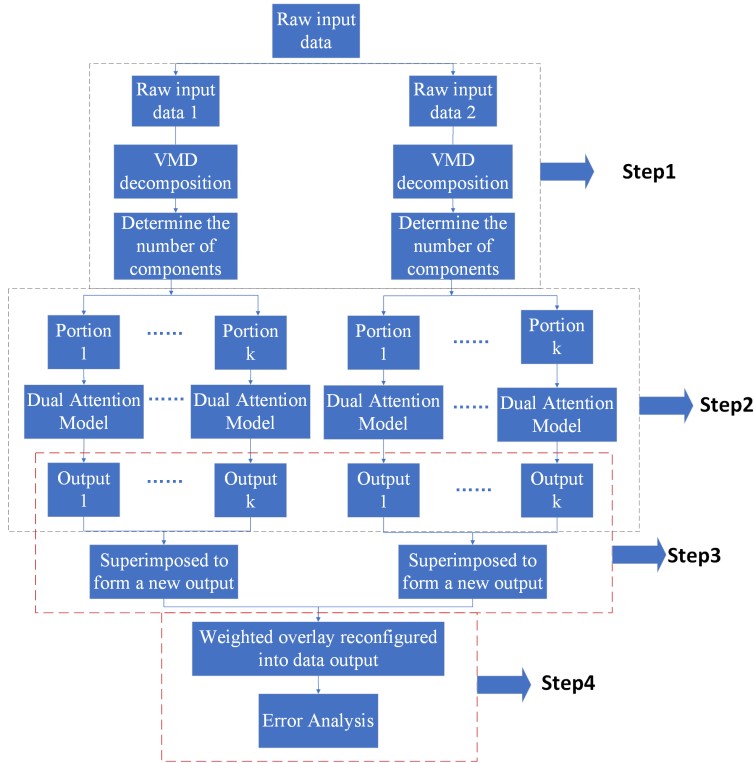

**Figure 1.** Power system timing data recovery model based on improved VMD and double radial GRU with a triple attention mechanism.

- The mode number k in the VMD technique is ascertained via the dual-threshold filtration method. Subsequently, the acquired k value and VMD methodology are employed to disintegrate the time-series data bilaterally, yielding k modal numbers.
- A CNN-GRU forecasting model is erected for each component, incorporating a dual attention mechanism. The input coding assay incorporates a feature attention mechanism to excavate the correlation between the time series data and the corresponding feature quantity. On the other hand, the output coding assay integrates a temporal attention mechanism to unearth the pertinent relationship between temporal data and missing data on time scales.
- The data of each component are superimposed and reconstructed to complete the data recovery results on one side.
- The data on either side of the absent data within the power system possess excellent completeness; hence, both sides of the missing data are individually modeled. Subsequently, the ultimate data restoration outcome is attained by incorporating the data restoration results of one side through adaptive weight allocation. The results are then scrutinized.

## 3. Decomposition of Quantitative Time Series Data Based on Improved VMD

VMD is an adaptive, quasi-orthogonal decomposition method as well as a more cutting-edge signal processing method that was proposed in 2014. The essence of the algorithm is Wiener filtering for noise reduction [20]. The goal is to decompose a time series data $X$ adaptively into several Intrinsic Mode Functions (IMFs) with finite bandwidth $x_k$. In order to calculate the spectral bandwidth of each component, this goal can be achieved by the following three steps: (1) Obtain the one-sided spectrum by Hilbert transform. (2) Transfer the spectrum of each quantitative data component to the baseband region by mixing an index tuned to the respective estimated center frequency. (3) Estimate the bandwidth of the decomposed signal by H1 Gaussian smoothing. For the input signal, the constrained variational model signal is represented as follows.

$$\min_{\{x_k\},\{\omega_k\}} \left\{ \sum_k \left\| \partial_t \left[ \left( \delta(t) + \frac{j}{\pi t} \right) * x_k(t) \right] e^{-j\omega_k t} \right\|_2^2 \right\}$$
$$\text{s.t.} \sum_k x_k = X \tag{1}$$

The center frequency of the corresponding component is represented by the letter $\{\omega_k\}$ in the equation above. The original input signal $X$ should be represented by the superposition of all components. Equation (1) can be changed into the following form by adding the quadratic penalty term and the Lagrange multiplier $\lambda$ to reconstruct the constraint:

$$\mathcal{L}(\{x_k\}, \{\omega_k\}, \{\lambda\}) = \alpha \sum_k \left\| \partial_t \left[ \left( \delta(t) + \frac{j}{\pi t} \right) * x_k(t) \right] e^{-j\omega_k t} \right\|_2^2 +$$
$$\left\| X(t) - \sum_k x_k(t) \right\|_2^2 + \left\langle \lambda(t), X(t) - \sum_k x_k(t) \right\rangle \tag{2}$$

where $\alpha$ is the penalty factor, which ensures the signal reconstruction's accuracy even when noise interference is present. The Lagrangian multiplier, $\lambda$, firmly guarantees that the constraints are upheld. The components and their center frequencies can be obtained from the saddle points of the above extended Lagrangian equation using the alternate direction method of multipliers (ADMM).

$$\hat{u}_k^{n+1}(\omega) = \frac{\hat{f}(\omega) - \sum_{i \neq k} \hat{u}_i(\omega) + \frac{\hat{\lambda}(\omega)}{2}}{1 + 2\alpha(\omega - \omega_k)^2} \tag{3}$$

$$\omega_k^{n+1} = \frac{\int_0^\infty \omega |\hat{u}_k(\omega)|^2 d\omega}{\int_0^\infty |\hat{u}_k(\omega)|^2 d\omega} \tag{4}$$

where $\hat{u}_k^{n+1}$ represents the residual of the current Wiener filter, and the frequency domain mode $\hat{u}_k(\omega)$ can be Fourier inverted to produce the time domain mode $x_k(t)$ before its real part is taken [21].

The final number of modes the VMD algorithm produces is determined by the modal number k, which has a non-negligible position in the algorithm. The center frequencies of the decomposed components are typically compared to determine whether there is under- or over-resolution, which has a relatively high degree of subjectivity. To a certain extent, the double-threshold screening method can prevent this issue [22].

The VMD analysis's components have narrow bandwidth properties, meaning that most of the modes are centered around the center frequency and have high corresponding amplitudes. Two thresholds—amplitude thresholds $T_2$ and and frequency interval thresholds $T_1$ —are established based on the aforementioned characteristics. By examining the spectral properties of the input timing signal and the threshold $T_1$, it is possible to divide the entire spectrum into several frequency bands, with each band being used as a potential

component. The frequency bands above are measured for their corresponding amplitude using the threshold $T_2$, and those whose amplitude satisfies the criteria are kept, while those with insufficient amplitude are ignored. The following four steps can be used to divide the application of the double threshold screening method to ascertain the modal number k: (1) Based on the spectral properties of the input data being examined, choose the appropriate frequency interval threshold $T_1$ and amplitude threshold $T_2$. (2) Look for local maxima in the spectrum, and using the frequency interval threshold $T_1$, divide the local maxima into corresponding frequency bands. (3) Examine the valid frequencies divided into each band; the valid frequency bands are those whose amplitudes meet the $T_2$ amplitude threshold. (4) The number of modes equals the number of legal frequency bands.

## 4. Dual Attention Model

Time series data of power systems are generated in chronological order. Considering the need for data analysis and storage, the power system timing data are often composed of discrete time series data. There is usually some connection between each power system timing data and other power time series data [23]. For the missing power system timing data, the reconstruction of power system data can be realized by analyzing the more complete time series data on both sides of it and mining the inner correspondence law between the complete power time series data and the missing power system timing data on the basis of considering the temporal sequence characteristics [24].

Figure 2 presents a comprehensive schematic of the dual attention model implementation, which comprises three primary components: the CNN layer, the GRU layer, and the attention layer. The components derived from the enhanced VMD algorithm are normalized using MinMaxScaler to confine the overall value range between 0 and 1 and subsequently fed into the CNN layer. During data input, the individual features of the input are weighted using the feature attention mechanism, thereby reinforcing the features that exert a significant impact on the outcome. On the output side, the temporal attention mechanism is employed to enhance the model accuracy by capitalizing on the correlation of the data on the time scale.

### 4.1. CNN-GRU Neural Network

Through a convolutional kernel, convolutional neural networks extract features from the input data locally, and convolutional neural network units learn the patterns in the input data window [25]. Convolutional neural networks gain two significant characteristics: First, the patterns studied by a convolutional neural network are translation invariant. A specific pattern learned by a convolutional neural network in a local data segment can identify this pattern in an arbitrary place.

Due to the stability of electricity consumption habits, the power system data possesses apparent repeatability. For example, the curve of regional load data in the same period of two days has substantial similarity, and the photovoltaic generation curve also shows strong similarity in two days with similar climate environments. This similarity provides a reasonable basis for convolutional neural networks. Second, the convolutional neural network can learn patterns' spatial hierarchy; the first convolutional layer can learn smaller local patterns, and the second convolutional layer further reconstructs the first convolutional layer's patterns to form larger patterns. This feature can make the convolutional neural network can learn more and more complex and abstract data [26]. Regarding temporal data processing, one-dimension (1D) convolution is commonly utilized. The operational principle of one-dimension convolution is depicted in Figure 3. The 1D convolutional layer is adept at detecting local patterns in a sequence. As the same input transformation is applied to each sequence segment, patterns discovered at one position in the temporal data can be subsequently recognized at other positions, rendering the 1D convolutional neural network translation invariant (concerning time translations).

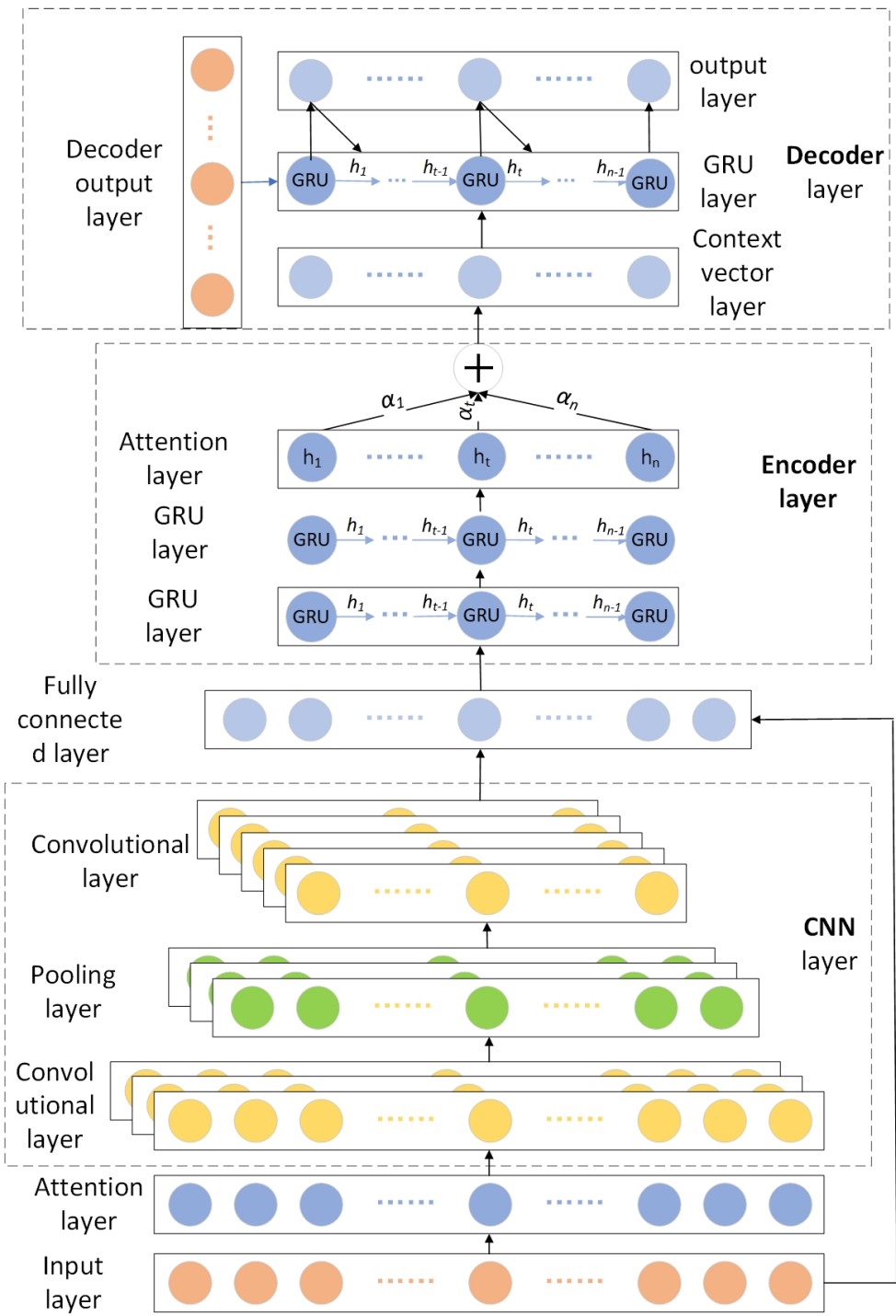

**Figure 2.** Dual attention CNN-GRU model.

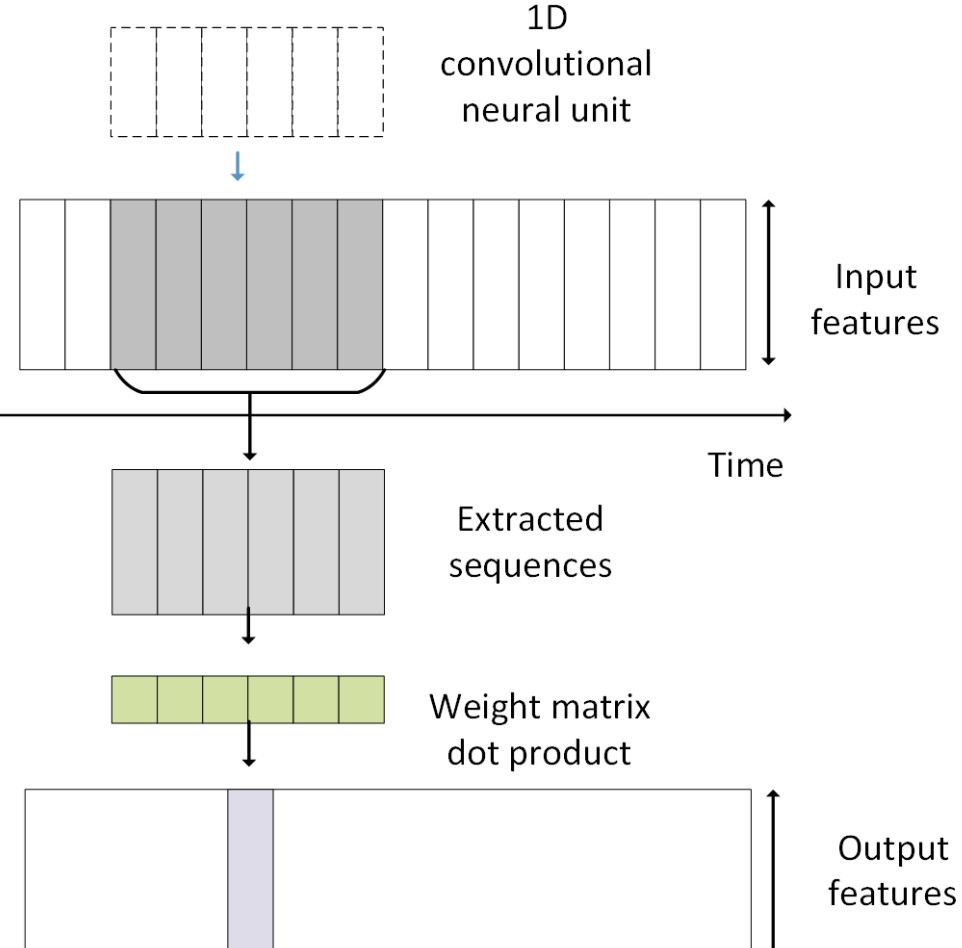

**Figure 3.** Working principle of one-dimensional convolution neural network.

By sampling the feature map through the maximum pooling layer, it can reduce the number of input features in the upper convolutional layer by sampling the input signal, which makes the model structure more streamlined, and the number of parameters to be computed decreases significantly. The maximum pooling layer extracts the essential information from the upper layer and transfers it to the lower convolutional layer, which allows the convolutional layer to have a larger and larger observation window (the window covers the proportion of the original input size), thus making the convolutional layer neural network have a spatial hierarchy.

A variant of the long short-term memory (LSTM) neural network, the GRU is a relatively new neural network structure primarily proposed by Junyoung Chung et al. at the International Conference on Machine Learning in 2015 [27]. In the GRU, the cell state and hidden state are combined with forgetting and input gates to create a single update gate. When applied to large-scale data, the original three gate structures are combined into just two gates, the parameters are decreased while maintaining the characteristics of LSTM, and the computational speed is thus noticeably increased.

The structure of the GRU shows the two gates that the GRU has: the reset gate rt and the update gate $Z_t$ [28]. The update gate is a linear transformation of the input signal $x_t$ at time step $t$ and the state $h_{t-1}$ at the previous time step, respectively, which are added together, and the information obtained is activated by a sigmoid function. An update gate determines how much of the signal from the past is going to be passed to the future. The reset gate is similar to the update gate in that the input signal $x_t$ at time step $t$ and the state $h_{t-1}$ at the previous time step are linearly transformed and added together. The resulting information is activated by a sigmoid function. However, the essence is

to decide how much information needs to be forgotten. After obtaining the reset gate, the linearly transformed reset gate and the linearly transformed input are added. The result is put into the hyperbolic tangent activation function to obtain the current required memory content $\hat{h}_t$. A unit of 1 is subtracted from the output of the update gate to obtain a difference. Then, the product of this difference and the current memory content is multiplied by the product of the state of the previous time step and the result of the update gate to obtain the state of the present time step. The specific formula for GRU is shown below.

$$
\begin{aligned}
z_t &= \sigma_g(W_z x_t + U_z h_{t-1} + b_z) \\
r_t &= \sigma_g(W_r x_t + U_r h_{t-1} + b_r) \\
\hat{h}_t &= \phi_h(W_h x_t + U_h(r_t \odot h_{t-1}) + b_h) \\
h_t &= (1 - z_t) \odot h_{t-1} + z_t \odot \hat{h}_t
\end{aligned}
\tag{5}
$$

where $x_t$ denotes the input vector; $h_t$ denotes the output vector; $\hat{h}_t$ denotes the current desired memory content; $z_t$ denotes the update gate; $r_t$ denotes the reset gate; $W$, $U$ and $b$ represent the parameter matrices and vectors; $\sigma_g$ denotes the sigmoid function; $\phi_h$ denotes the hyperbolic tangent function; and $\odot$ denotes the Hadamard product.

*4.2. Attentional Mechanisms*

The processing of visual signals by attentional mechanisms is specific to human vision. Human vision requires daily access to enormous amounts of image data as one of the most crucial information acquisition channels. The brain typically concentrates on the image's critical components while ignoring the image's comparatively minor components when processing this information. This mechanism has the potential to speed up and improve the processing of visual data significantly [29]. The attention mechanism in this paper is similar to this in that the important parts of the signal are selected and given a relatively large weight, resulting in a greater increase in the output accuracy of the whole system.

Let the input timing and the corresponding characteristics be:

$$
x = [x_1, x_2, \dots, x_T] = [x^{(1)}, x^{(2)}, \dots, x^{(n)}]^T
\tag{6}
$$

The expansion can be represented by the following matrix.

$$
x = \begin{bmatrix}
x_1^{(1)} & x_1^{(2)} & \cdots & x_1^{(n)} \\
x_2^{(1)} & x_2^{(2)} & \cdots & x_2^{(n)} \\
\vdots & \vdots & & \vdots \\
x_T^{(1)} & x_T^{(2)} & \cdots & x_T^{(n)}
\end{bmatrix} \in \mathbf{R}^{T \times n}
\tag{7}
$$

$x_t = \left[x_t^{(1)}, x_t^{(2)}, \cdots, x_t^{(n)}\right] (1 \le t \le T)$ is the feature set of the above n features at moment $t$. $x^{(m)} = \left[x_1^{(m)}, x_2^{(m)}, \cdots, x_T^{(m)}\right] (1 \le m \le n)$ is the value of the mth relevant eigenvalue at moment $t(1 \le t \le T)$.

In order to obtain the association of each feature variable with the current time series, i.e., for the present time series, the importance of its corresponding feature quantity, the feature attention method is used for calculation. The attention weights corresponding to the feature quantities at the current moment are calculated by associating the feature variables at the moment $t$ with the corresponding output variables at moment $t - 1$, $h_{t-1}$ and the state variables, $s_{t-1}$.

$$
e_t^{(m)} = V_e^{\mathrm{T}} \tanh\left(W_e[h_{t-1}; s_{t-1}] + U_e x^{(m)} + b_e\right)
\tag{8}
$$

$V_e, W_e, U_e$ are the weight matrices, respectively. $b_e$ is the corresponding bias term.

After obtaining the weight values, they need to be normalized so that the sum of the weight values corresponding to moment $t$ is 1.

$$\alpha_t^{(m)} = \frac{\exp\left(e_t^{(m)}\right)}{\sum_{i=1}^{n} \exp\left(e_t^{(i)}\right)} \tag{9}$$

After obtaining the weighting coefficients, the input eigenvalues are multiplied by them to obtain the weighted input:

$$\widetilde{x}_t = \left[\alpha_t^{(1)} x_t^{(1)}, \alpha_t^{(2)} x_t^{(2)}, \cdots, \alpha_t^{(n)} x_t^{(n)}\right] \tag{10}$$

The obtained adaptive weighted input $\widetilde{x}_t$ afterward is fed into the subsequent model instead of the original input $x_t$. This method can dynamically extract the correlation between the feature values and the corresponding time series. The state $h_t$ of the hidden layer at each moment needs to be updated at the next moment.

$$h_t = f_1(h_{t-1}, \tilde{x}_t) \tag{11}$$

$f_1$ is the GRU network unit.

After the adaptive feature attention results are computed, the output results obtained by the feature attention mechanism are used as the input of the temporal attention mechanism in the next stage. The time series attention mechanism focuses attention on the input series and obtains the adaptive time principal series output by weighted average for the input time series [30]. Figure 4 illustrates the principle of the attention mechanism implementation. The attentional mechanism is calculated as follows:

$$c_i = \sum_{s=1}^{T} \alpha_{ts} \overline{h}_s \tag{12}$$

$$\alpha_{ts} = \frac{\exp\left(score\left(h_t, \overline{h}_s\right)\right)}{\sum_{s'=1}^{S} \exp\left(score\left(h_t, \overline{h}_{s'}\right)\right)} \tag{13}$$

where $h_t$ is the output of the decoder corresponding to time $t$ and $\overset{h}{h} -_s$ is the source hidden state of the encoder. Here, score expression is specifically calculated as follows:

$$score\left(h_t, \overline{h}_s\right) = h_t^{\top} \overline{h}_s \tag{14}$$

The core idea is to make the context vector $c_t$, which is otherwise invariant in the seq2seq structure, dynamic by reorganizing it several times. The traditional context vector $c_t$ selects the output of the last time step as the final output because the data can be regarded as the process of gradually extracting features after GRU processing. The output of the last time step often contains important information about the state of the past time steps. However, for data with long time steps and strong information correlation, this model often leads to the vital information of a past time step being ignored or not highlighting the information strongly associated with the prediction results [31]. Through the temporal attention mechanism, the vector $c_t$ is dynamized, and $c_t$ is no longer just the output of the last time step but also a dynamic combination of individual time steps. Different weight coefficients are given for different predicted contents, thus giving different $c_t$, which finally achieves the role of extracting necessary information in the time series.

Fuse the real-time updated context vector $c_t$ with the output $h_t$ of time step $t$ as the input of the decoder:

$$\tilde{h}_t = \tanh(W_c[c_t; h_t] + b_c) \tag{15}$$

where $W_c$ and $b_c$ are the weight and bias of the fused input, respectively; tanh is the hyperbolic tangent function; $\tilde{h}_t$ is the encoder output.

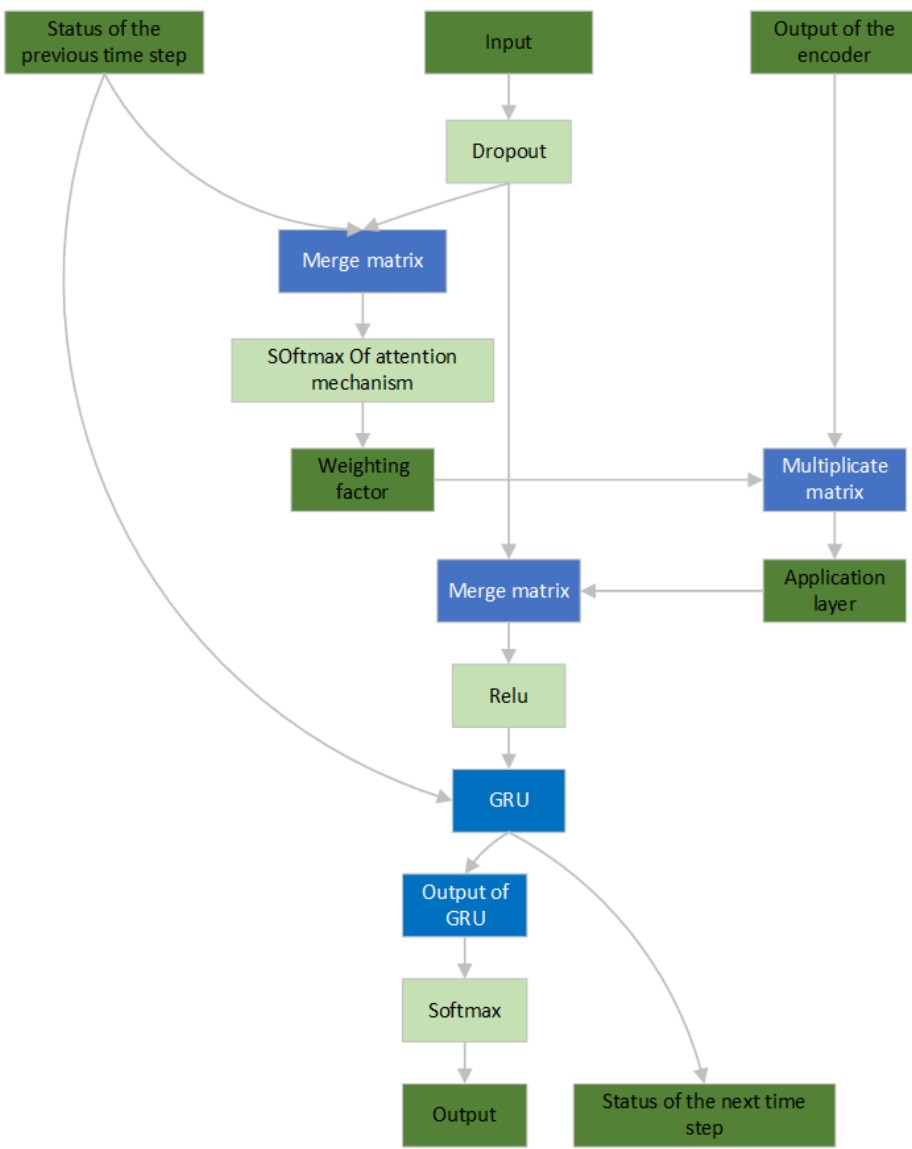

**Figure 4.** Schematic diagram of realization of attention mechanism.

## 5. Example Analysis

### 5.1. Data Pre-Processing and Error Indicators

In this paper, three datasets are used to validate the model proposed in this paper: Singapore power load data, a public wind power dataset in the United States, and a public photovoltaic (PV) output dataset in Australia. The first 80% of the data is used as the training set, the first 80% to 95% is used as the validation set, and the last 5% is used as the test set. The wind power generation data set has a total of 50,500 points, with a sampling period of 10 min. Again, the first 80% of the data is used as the training set, the first 80% to 95% of the data is used as the validation set, and the last 5% of the data set is used as the test set. The photovoltaic output at night is always zero, so there is no meaningful training for this period, and the photovoltaic data in the night part of the data have been removed during the data processing. The photovoltaic output data set has 98,000 points with a sampling period of 5 min. Here, 80% of the data is used as the training set, the first 80% to 95% is used as the validation set, and the last 5% is used as the test set. In order to make the neural network training more efficient beforehand, the MinMaxScaler in sklearn was used to normalize the data by min–max between (0, 1), which are calculated as:

$$MAE = \frac{1}{m} \sum_{n}^{i=1} |(y_i - \hat{y}_i)| \tag{16}$$

$$MSE = \frac{1}{m} \sum_{m}^{i=1} (y_i - \hat{y}_i)^2 \tag{17}$$

where $m$ denotes the total number of model output results; $y_i$ and $\hat{y}_i$ mean the actual value of the $i$th point in the output results and the model output value, respectively. Smaller MAE and MSE indicate better model fit.

Adam is chosen as the optimizer (adaptive moment estimation), and the Adam optimization algorithm can achieve adaptive gradient selection, which can jump out some local minima.

### 5.2. Model Configuration

The convolutional layers are stacked in two layers. The number of neurons in each layer is 128, the convolutional window is 64, and the activation function is selected as Relu. The pooling layer is selected as the maximum pooling. The encoder chooses two layers of GRU stacking: 128 neurons for the first layer and 64 neurons for the second layer. Tanh is selected for both activation functions, the discard rate is set to 0.1 for the first layer and 0.24 for the second layer, and the decoder chooses single-step GRU decoding with 64 neurons. Each GRU step is output through a unit fully connected layer. The training step size is 0.0075.

### 5.3. Data Set Comparison

In this paper, a total of five models are introduced for comparison, and the five models are LSTM, multi-layer perceptron (MLP), Seq2Seq, CNN and the model of this paper using one-sided reconstruction. All models use the same normalization procedure for data output, and the input of the models is the corresponding historical sequence data to ensure the scientific accuracy and validity of the comparison method. The data in the test set were taken from 32 sampling points and 128 sampling points for data reconstruction, and the evaluation results of each quantitative data index are shown in Tables 1 and 2. In order to make the comparison results more intuitive and comparable, the comparison results are directly normalized using the model output results, which can avoid the problem of the inability to compare because of different units and large differences in the size of the original data.

**Table 1.** MAE and MSE of each model with reconstructed length of 32 sampling points.

|  |  | Load Data Set | Wind Power Dataset | Photovoltaic Power Dataset |
|---|---|---|---|---|
| Model of this paper | MAE/(e-2) | 1.96 | 5.38 | 4.48 |
|  | MSE/(e-4) | 5.97 | 37.29 | 31 |
| One-sided modeling | MAE/(e-2) | 2.9 | 9.58 | 8.28 |
|  | MSE/(e-4) | 11.6 | 74.06 | 61.7 |
| LSTM | MAE/(e-2) | 2.27 | 8.25 | 6.87 |
|  | MSE/(e-4) | 7.49 | 61.68 | 51.4 |
| CNN | MAE/(e-2) | 2.48 | 8.54 | 7.12 |
|  | MSE/(e-4) | 8.57 | 65.04 | 54.2 |
| Seq2seq | MAE/(e-2) | 2.14 | 7.12 | 5.84 |
|  | MSE/(e-4) | 6.57 | 44.09 | 36.7 |
| MLP | MAE/(e-2) | 2.85 | 8.54 | 7.21 |
|  | MSE/(e-4) | 7.98 | 59.46 | 49.8 |

**Table 2.** MAE and MSE of each model with reconstructed length of 128 sampling points.

| Model of This Paper | | Load Data Set | Wind Power Dataset | Photovoltaic Power Dataset |
|---|---|---|---|---|
| One-sided modeling | MAE/(e-2) | 2.74 | 9.95 | 6.97 |
| | MSE/(e-4) | 11.14 | 83.53 | 64.48 |
| LSTM | MAE/(e-2) | 4.09 | 16.77 | 13.33 |
| | MSE/(e-4) | 21.46 | 165.89 | 125.25 |
| CNN CNN | MAE/(e-2) | 3.28 | 20.54 | 10.85 |
| | MSE/(e-4) | 15.22 | 108.56 | 114.11 |
| Seq2seq | MAE/(e-2) | 3.59 | 21.26 | 10.96 |
| | MSE/(e-4) | 17.22 | 101.46 | 123.58 |
| MLP | MAE/(e-2) | 3.08 | 12.67 | 9.46 |
| | MSE/(e-4) | 13.25 | 106.7 | 81.11 |
| | MAE/(e-2) | 4.29 | 15.79 | 11.82 |

5.3.1. Reconstructed Data Length of 32 Sampling Time Points Reconstruction Effect Comparison

Table 1 presents a summary of the Mean Squared Error (MSE) and Mean Absolute Error (MAE) for various models with a reconstructed data length of 32. The experimental results from three data sets, namely, the load data set, wind power generation data set, and photovoltaic power generation data set, are arranged from left to right. As the summary table employs normalized data results, it does not include any units. Figures 5–7 depict the schematic diagram of the load data reconstruction results, wind power generation data reconstruction results, and photovoltaic power generation data reconstruction results, respectively, with a reconstructed data length of 32 sampling time points. To facilitate comparison and highlight the characteristics of model reconstruction, a period with more salient features was selected from each of the three datasets.

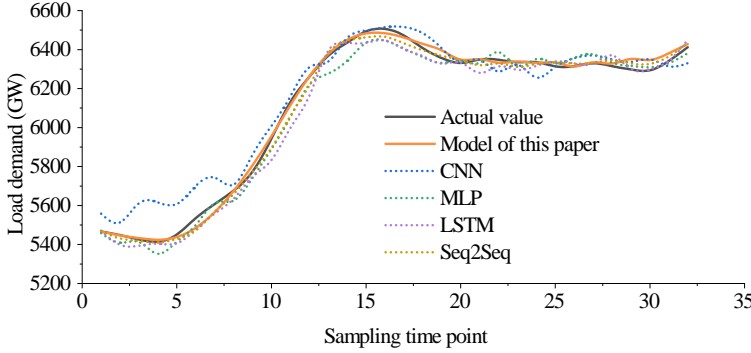

**Figure 5.** Results of reconstructing 32 load time sampling points with different models.

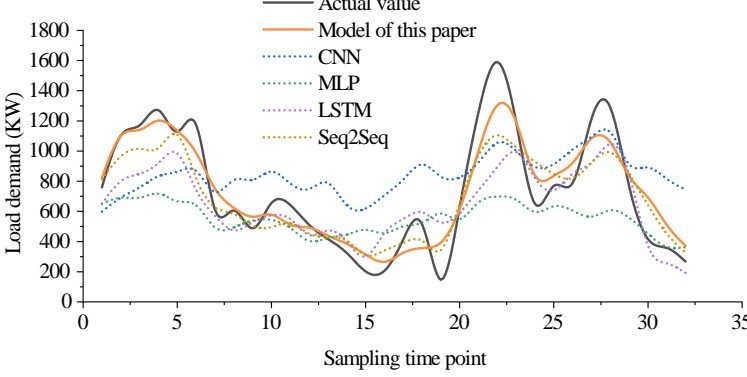

**Figure 6.** Results of reconstructing 32 wind power time sampling points with different models.

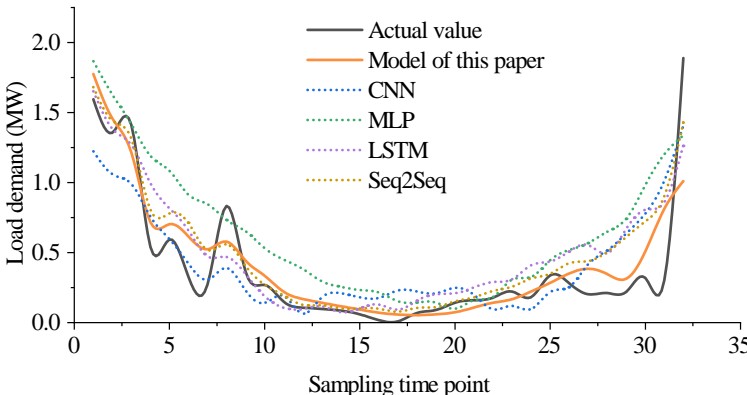

**Figure 7.** Results of reconstructing 32 photovoltaic power time sampling points with different models.

For the loaded data set, we can see that when the reconstructed data length is 32 sampling time points, the average MAE of the proposed model in this paper decreases by 32.41%, 13.66%, 20.97%, 8.41%, and 31.23% relative to the unilateral reconstruction model, LSTM, CNN, Seq2Seq, and MLP, respectively. The mean MSE of the proposed model decreases by 48.53%, 20.29%, 30.34%, 9.13%, and 25.19% relative to the one-sided reconstruction model, LSTM, CNN, Seq2Seq, and MLP, respectively.

For the wind power dataset, we can see that when the reconstructed data length is 32 samples, the mean MAE and mean MSE of the model proposed in this paper decreased by 43.83%, 34.79%, 37.01%, 24.44%, 15.42%, 37.00%, and 49.65%, 39.54%, 42.67%, 15.42%, and 37.29%, respectively.

For the photovoltaic generation dataset, we can see that when the reconstructed data length is 32 samples, the MAE mean and MSE mean of the proposed model in this paper decreased by 45.89%, 34.79%, 37.08%, 23.29%, 37.86%, and 49.76%, respectively, when compared to the unilateral reconstruction model, LSTM, CNN, Seq2Seq, and MLP. 39.69%, 42.80%, 15.53%, and 37.75%.

The reconstructed data results graph demonstrates that the model proposed in this paper is more closely aligned with the actual trend for data reconstruction. Specifically, the proposed model better reconstructs the data mutation part. An extreme value point near the fifteenth sampling point is observed for the load reconstruction results, and the proposed model fits this extreme value point better. Furthermore, the rise and fall of the load reconstruction results are also in close proximity to the actual data. The effect is more pronounced for the photovoltaic power generation data and wind power generation data with apparent changes. The wind power data reconstruction results show that the data are more fluctuating, and the data are relatively very variable, with changes occurring repeatedly within 32 sampling points. The proposed model can react quickly at the abrupt change points, such as the 6th sampling point, 19th sampling point, 22nd sampling point, and 28th sampling point, and keep up with the data change trend in time. The relative change of photovoltaic data is located between the load and wind power generation. However, photovoltaic data have an obvious characteristic that the data will occasionally suddenly drop to very low or rise from very low to a higher position. This characteristic is mainly due to the photovoltaic power generation's dependence on weather, which has a significant impact on light intensity. For all three data sets in this paper, the proposed model's reconstructive ability on relatively short missing data is significantly stronger than the comparison model.

### 5.3.2. Reconstructed Data Length of 128 Sampling Time Points Reconstruction Effect Comparison

Table 2 presents a summary of the MSE and MAE for each model with a reconstructed data length of 128. It includes experimental results from three datasets. Appendix A Figure A1 displays a schematic diagram of the reconstructed load data with a reconstructed data length of 128 sampling time points. Appendix A Figure A2 exhibits a schematic diagram of the

reconstructed wind power generation data with a reconstructed data length of 128 sampling time points. Appendix A Figure A3 illustrates a schematic diagram of the reconstructed photovoltaic power generation data with a reconstructed data length of 128 sampling time points. The three datasets were selected for their distinct performance characteristics over a period to facilitate comparison and emphasize the features of model reconstruction.

For the loaded dataset, when the reconstructed data length is 128 samples, the mean MAE of the model proposed in this paper decreases by 33.01%, 16.56%, 23.68%, 11.04%, and 36.13% relative to the one-sided reconstruction model, LSTM, CNN, Seq2Seq, and MLP, respectively. The mean MSE of the model proposed in this paper decreased by 48.09%, 26.81%, 35.31%, 15.92%, and 37.49% relative to the one-sided reconstruction model, LSTM, CNN, Seq2Seq, and MLP, respectively.

For the wind power dataset, when the reconstructed data length is 128 samples, the mean MAE and mean MSE of the model proposed in this paper decreased by 40.67%, 51.56%, 53.20%, 21.47%, 36.99%, and 49.65%, respectively, when compared to the unilateral reconstruction model, LSTM, CNN, Seq2Seq, and MLP. 23.06%, 17.67%, 21.72%, and 44.69%.

For the photovoltaic generation data set, when the reconstructed data length is 128 samples, the mean MAE and mean MSE of the model proposed in this paper decreased by 47.71%, 35.76%, 36.41%, 26.31%, 41.03%, and 48.52%, respectively, when compared to the unilateral reconstruction model, LSTM, CNN, Seq2Seq, and MLP, generating values of 43.49%, 47.82%, 20.50%, 47.79%.

We observe that the model proposed in this paper more closely approximates the actual trend in data reconstruction. Specifically, in the case of data mutations, the model presented here performs better. Similar to the analysis with 32 samples, in the load reconstruction graph, although the change structure of 128-sample data is more prominent than that of 32-sample data, our model can track changes more effectively. Likewise, this model outperforms other models in tracking wind and photovoltaic reconstructed graphs. As with the analysis of 32-sample reconstructions, our model excels at handling large data changes over short periods of time in wind reconstruction; for instance, it fits more closely around the 40th and 60th sample points. The same conclusion applies to photovoltaic data reconstruction. It is evident that our model also surpasses other models in reconstructing data at 128 sampling time points. For all three datasets examined in this paper, our proposed model significantly outperforms comparison models in reconstructing relatively long missing data.

### 5.3.3. Reconstructing Data Error Distribution Analysis

Figures 8–10 display error analysis plots for the reconstruction results of our model and other comparison models at 32 sampling time points. Appendix A Figures A4–A6 present error analysis plots for the reconstruction results of our model and other comparison models at 128 sampling time points. From the perspective of error for different lengths of missing data, the range and median error of the error distribution for 128 sampling time points in all three datasets are significantly higher than those for 32 sampling time points. For instance, the median absolute values of load errors for our model, CNN, and MLP increase from 0.07, 0.126, and 0.25 to 0.31, 0.49, and 0.55, respectively. In terms of absolute error value distribution data, the reconstruction results of our proposed model are more concentrated around the median compared to several other models.

Figure 8 and Appendix A Figure A4 display the error plots of load data reconstruction results for 32 and 128 sampling points, respectively. The analysis of the load data reveals that the reconstruction error of 128 sampling points for all models is significantly larger than the error of 32 sampling points. For the proposed model, most of the errors increase from the 0–50 GW interval to the 0–115 GW interval. However, compared to other models, the error distribution of the proposed model is still relatively concentrated, which is consistent with the conclusion that the MAE and MSE are relatively smaller in the previous analysis. The performance of the proposed model is more outstanding in the error distribution interval 10–90%, error distribution interval 25–75%, and median error

distribution, especially the 25–75% error distribution index, which is lower than other models. This indicates that most of the errors of the proposed model are concentrated around 0. Compared to other models, the main reason for the superior performance of the CNN model in the 32 sampling case is related to its working principle. CNN itself does not have the concept of time flow and uses the form of convolutional kernel to identify data features. Even if the recognition window is increased, as long as the features identified by the CNN convolutional kernel can still be applied, the error distribution of the CNN model does not change much.

Figure 9 and Appendix A Figure A5 depict error plots of wind power generation data reconstruction outcomes for 32 and 128 sampling points, respectively. Figure 10 and Appendix A Figure A6 portray error plots of reconstruction outcomes for photovoltaic generation data with 32 and 128 sampling points, respectively. In both cases, wind power data and photovoltaic data, we observe a similar phenomenon to the load data, where the error distribution of each model increases significantly. However, the model proposed in this paper is more dispersed and concentrated in comparison to the other models both for the 32-sample-point reconstruction and the 128-sample-point reconstruction. With regard to the 25–75% error distribution metric, it is evident that the proposed model outperforms other models.

Nonetheless, there are also some individual characteristics. For the wind power data, the rise in error distribution from 32 to 128 sampling points is not as evident as that of the load data set. This is primarily because the wind power data set is more intricate than the load data set, and the changing trend is more challenging to capture. Therefore, even if it is extended to 128 sampling points, the increase in error is limited, given that the error of 32 sampling points is already significant. As for the photovoltaic data, the error variation is between the load data set and the wind power data set, owing to its inherent regularity. However, there are some random variations in the data set.

The absolute value of Seq2Seq error in reconstructing the 128 data sampling points of the model load data set in this paper appears to be more concentrated than the model proposed in this paper. This is probably due to the load data set used being the load data of Singapore, which is relatively stable, and the trend changes are relatively evident. In addition, the data set has fewer dramatic fluctuations at certain points in time, making the data itself very reconfigurable. For the Seq2Seq model, this can be more perfect in grasping the data structure characteristics, and even without adding additional data processing means, the data reconfiguration effect is already excellent. However, for wind power generation data and photovoltaic power generation data with significant fluctuations and non-obvious trend changes, it has been challenging for the Seq2Seq model to capture the complete intrinsic pattern of the data, resulting in a significantly enlarged error distribution.

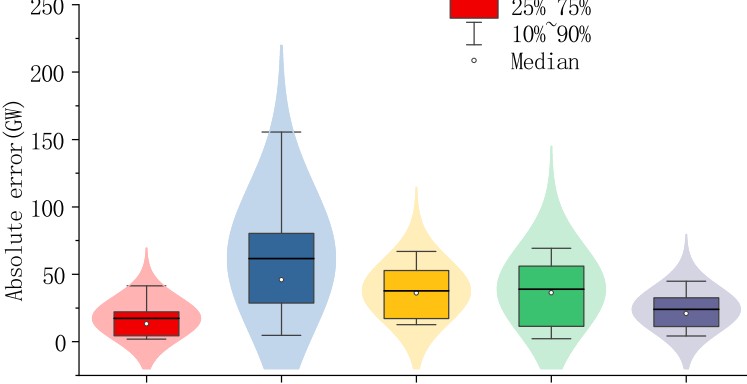

**Figure 8.** Error results of reconstructing 32 time sampling points of load with different models.

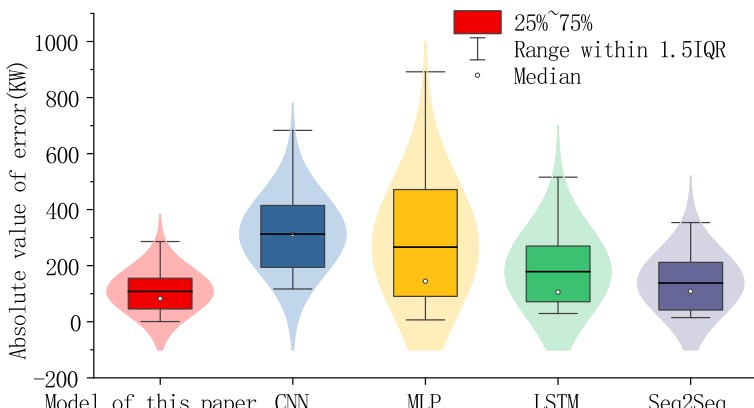

**Figure 9.** Error results of reconstructing 32 wind power generation time sampling points with different models.

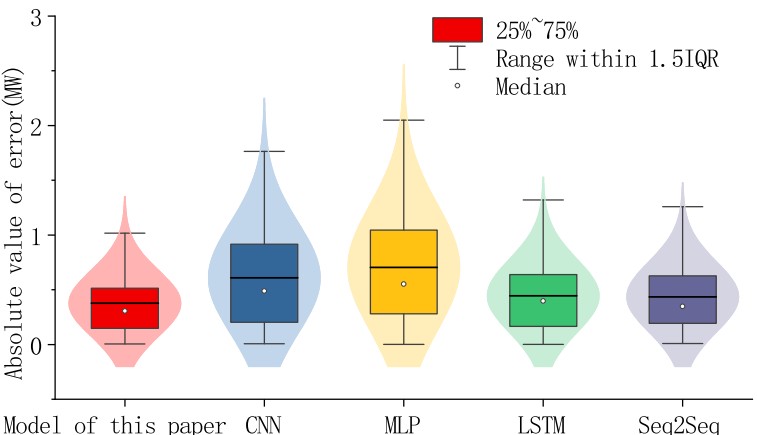

**Figure 10.** Error results of reconstructing 32 photovoltaic power generation time sampling points with different models.

5.3.4. 128 Sample Points Reconstruction Results Compared to 32 Sample Points Reconstruction Results

It is more challenging to reconstruct 128 sampled time points of data than to reconstruct 32 sampled time points of data. Comparing Tables 1 and 2, we can observe a significant increase in the MAE and MSE metrics for both models. This is because (1) the same data set is evidently less rich for 128 sampling points than for 32 sampling points, resulting in a model that is less comfortable with 128 sampling points. (2). Reconstructing 128 sampling points using the same deep learning model may lead to model limitations, as the model may not be able to handle more data points or capture more complex relationships in the data set, resulting in higher MAE and MSE metrics.

Figure 11 illustrates the percentage of MAE increase for 128 sample points compared to 32 sample points. Figure 12 illustrates the percentage increase of MSE for 128 samples compared to 32 samples. From Figure 11, it is evident that the MAE of each model increased significantly for 128 samples compared with 32 samples, and the most significant increase was 150% for LSTM and CNN. Furthermore, by comparing the three data sets, the MAE of this model is the smallest in most cases, indicating that this model makes the best use of the data. Similarly, it can be observed from Figure 12 that the performance of this model is superior in the other two datasets, except for the wind power dataset. The possible reasons for the low MSE of CNN and LSTM models in wind power data are: (1) The MSE of these two models is already relatively large in the province at 32 sampling points, which means that the model itself does not utilize the data to a high degree. (2) The regularity of the wind power data set itself is more difficult to find, which leads to the model itself not

resolving the data set well, resulting in the reconstruction length. When the length of the reconstruction increases, there is a more obvious decrease in the degree of utilization.

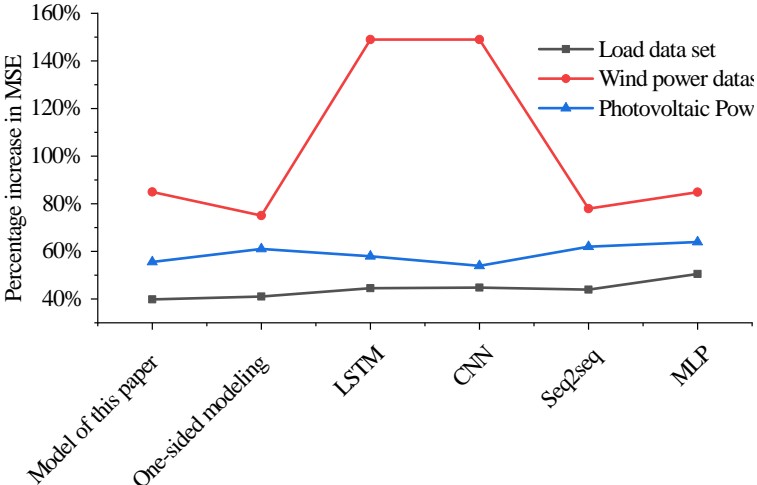

**Figure 11.** Percentage increase in MAE at 128 sampling points compared to 32 sampling points.

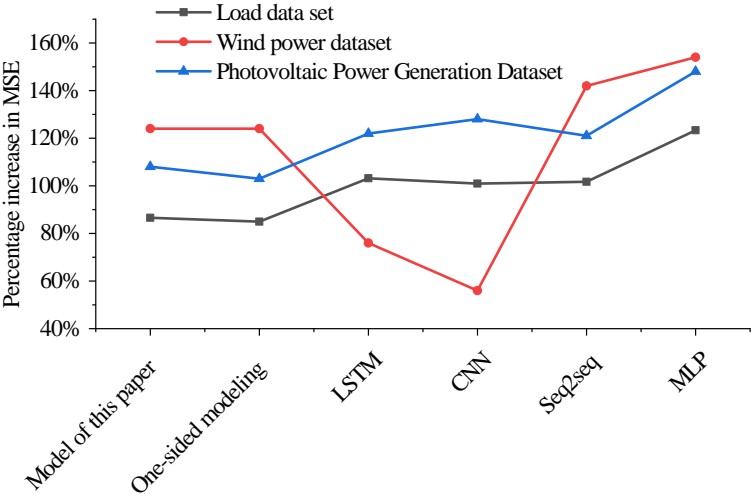

**Figure 12.** Percentage increase in MSE at 128 sampling points compared to 32 sampling points.

By comparing the load with 32 sampling time points and 128 sampling time points, it is evident that the prediction model in this paper has a better fit to the true value, and the accuracy decreases more slowly as the data length increases. Compared with other models, the model proposed in this paper not only reconstructs the trend of the missing data more accurately but also performs better for the part of the load with large fluctuations and can analyze the law of sudden changes to a greater extent, which improves the overall model accuracy. Comparing the wind power generation data and photovoltaic power generation data, which are more random and fluctuating, it is clear that the model proposed in this paper fits the sudden changes in the data significantly better than the other models, and it not only fits the changes in the trend of the data but also captures the drastic local changes more accurately, making the output of this model closer to the actual situation. For example, in the case of photovoltaic power generation data, although the overall trend can be captured more obviously, the overall data have the characteristic of fluctuating sharply up and down around the trend, and the model in this paper can capture the larger fluctuations more accurately, so that the details can be supplemented to make the overall error smaller.

## 6. Conclusions

This paper proposes a method to recover power system timing data based on improved VMD and attention mechanism bi-directional CNN-GRU, which initially processes the data by VMD so that the data can be better divided into multiple groups based on frequency centroids, extracts the temporal characteristics of the time series data by using a CNN model and then realizes the Seq2Seq structural model combined with multiple attention mechanisms for the reconstruction of the data. This paper compares the unilateral reconstruction model with LSTM, CNN, Seq2Seq, and MLP. It analyzes the characteristics of the model proposed in this paper compared to other models in terms of three indicators: MAE, MSE, and reconstruction result error. Although the result analysis is sensitive to the data scenarios, the following conclusions can be drawn:

- The model in this paper has strong data reconstruction capability, especially for fitting data mutations.
- This model has better performance in long time series data reconstruction compared to other models.
- This model has a more outstanding effect on data reconstruction with drastic changes compared to other models in this paper and has broader application potential.

In the future, we can study the characteristics of different models with a different focus on data analysis and combine multiple models dynamically to achieve more accurate data processing capability.

**Author Contributions:** Methodology, K.X.; Formal analysis, Y.L.; Resources, K.X.; Writing – original draft, K.X.; Writing– review and editing, J.L. All authors have read and agreed to the published version of the manuscript.

**Funding:** This research received no external funding.

**Conflicts of Interest:** The authors declare no conflict of interest.

## Appendix A

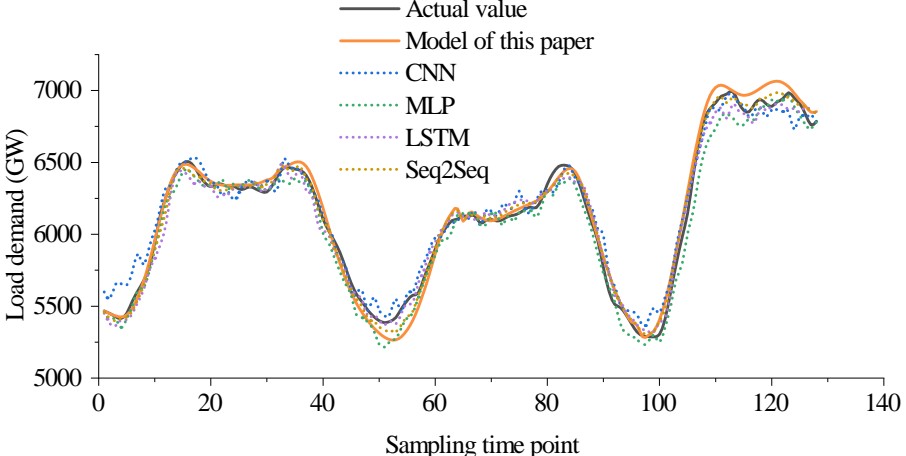

**Figure A1.** Results of reconstructing 128 load time sampling points with different models.

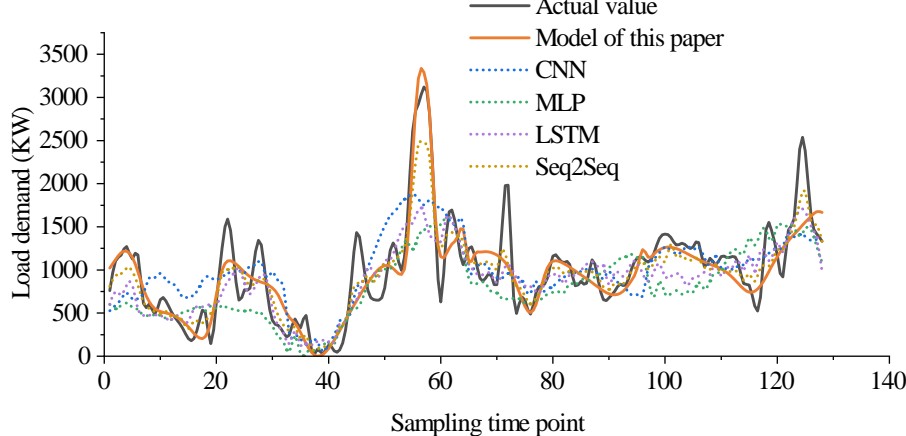

**Figure A2.** Results of reconstructing 128 wind power time sampling points with different models.

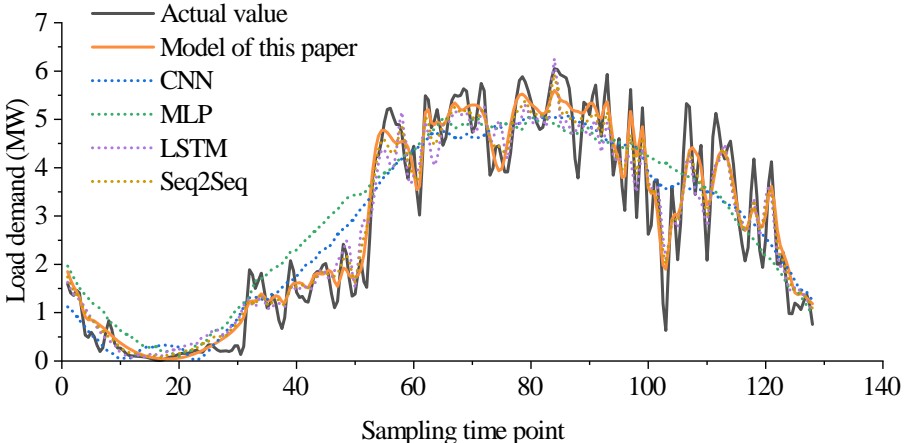

**Figure A3.** Results of reconstructing 128 temporal sampling points of photovoltaic power generation with different models.

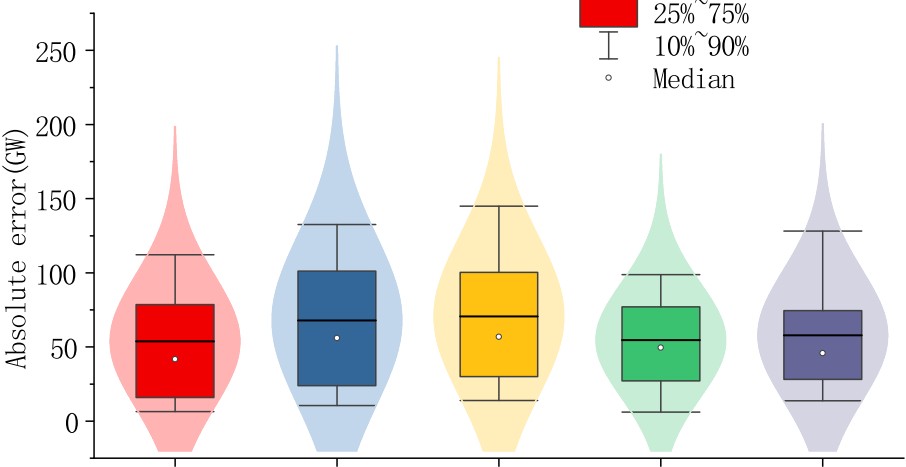

**Figure A4.** Error results of reconstructing 128 time sampling points of load with different models.

**Table A1.** Similar work summary table.

| Literature | Main Technical Means | Advantage | Disadvantages |
|---|---|---|---|
| [12] | A new method for detecting bad data in power systems using temporal correlation and statistical consistency of measurements is proposed. The method uses three innovative matrices to capture measurement correlation and statistical consistency, and it applies projection statistics to detect bad data. | The computational requirements are not large, and the data categories with high similarity in the power system can be more effectively detected and reconstructed for bad data. | The reconstruction is less effective for data with high dimensionality and complexity, complex change trends, and difficulty in finding patterns. |
| [13] | In the case of a small amount of synchronous phase volume data missing and using a Lagrangian interpolation polynomial approach to adaptively estimate incomplete and missing data. | Fast computation, practical, good for reconstructing data with small amount of 1D data. | The method has a small range and cannot be used once the missing data becomes long or the data are not one-dimensional. |
| [14–16] | The core idea is to take advantage of the similarity between the data column where the missing data are located and the complete data column and use this similarity to reconstruct the data through further data processing. | The performance is highly correlated with the degree of data similarity, and the deeper the similarity between the data, the better the method refactoring and vice versa. Supported by explicit mathematical principles. | Large differences between data can seriously affect the effectiveness of data reconstruction, and the larger the amount of data, the more patterns in the data and performance degradation. |
| [17] | A shallow coder is used to learn the data features and to complement the data by the data structure after the weighting process. | It also has better adaptability for complex data and has a more outstanding reconfiguration effect. | Poor reconfiguration for complex, variable data. |
| [18,19] | Ref. [18] learns the features of temporal data through a positive and negative GRU network. Ref. [19] then learn the features of the data using generative adversarial networks. | It also has better adaptability for complex data and has a more outstanding reconfiguration effect. | The mathematical mechanism is not clear. The algorithms take longer to compute and are more demanding on computational resources. |

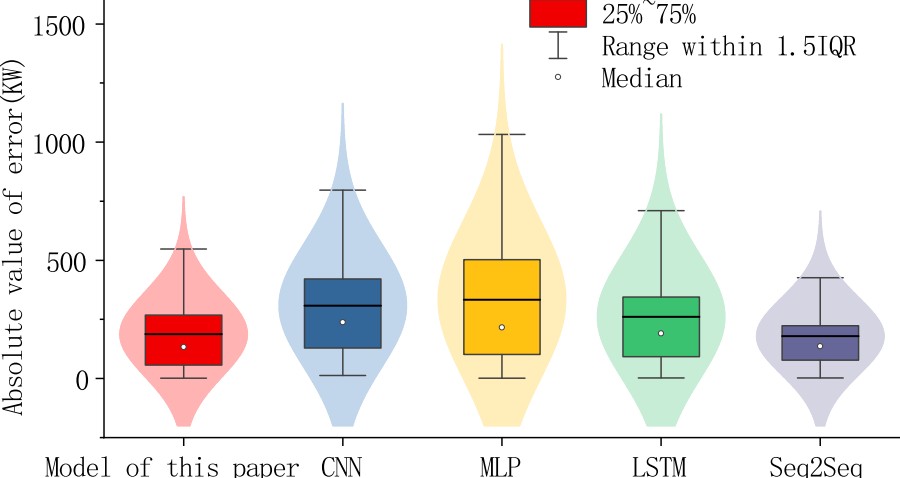

**Figure A5.** Error results of reconstructing 128 wind power generation time sampling points with different models.

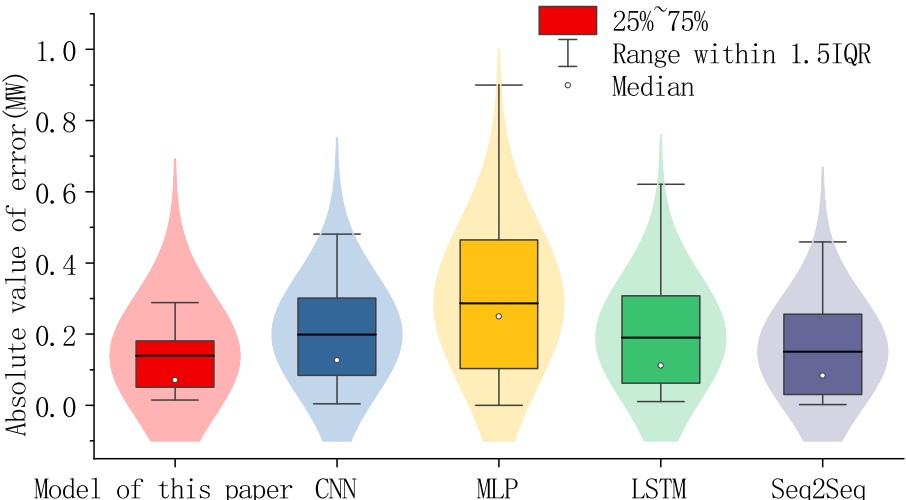

**Figure A6.** Error results of reconstructing 128 photovoltaic power generation time sampling points with different models.

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
