# Peer review of "A Power System Timing Data Recovery Method Based on Improved VMD and Attention Mechanism Bi-Directional CNN-GRU"

_electronics, doi:10.3390/electronics12071590_

Round 1
Reviewer 1 Report
The manuscript entitled "A Power System Measurement Data Recovery Method Based on Improved VMD and Attention Mechanism Bi-directional CNN-GRU" has been prepared by the authors. There are many ambiguities. Note the following;
1- The manuscript must be restructured.
2- It lacks innovation. The mentioned issue is a data recovery method for big data. It can be applied for all kind of data.
Author Response
Response to Reviewer 1 Comments
Dear Editor and Reviewers,
We appreciate you for your precious time in reviewing our paper and providing valuable comments. It was your valuable and insightful comments that led to possible improvements in the current version. The authors have carefully considered the comments and tried our best to address every one of them. We hope the manuscript after careful revisions meet your high standards. The authors welcome further constructive comments if any. Below we provide the point-by-point responses. All modifications in the manuscript have been highlighted in red.
Statement of Revision
Point 1: The manuscript entitled "A Power System Measurement Data Recovery Method Based on Improved VMD and Attention Mechanism Bi-directional CNN-GRU" has been prepared by the authors. There are many ambiguities. Note the following; 1- The manuscript must be restructured.
Response 1: Thank you for your feedback. We have revised the six dimensions of your journal's scoring index in response to your comments.
(a): Literature review: We thoroughly reviewed our cited literature, removing irrelevant sources and incorporating recent and relevant literature. For example, we have removed references 1, 2, and 4, and added new references that align more closely with the content of our article. We have also included references to related articles, such as GRU content and attention mechanisms, where appropriate. Overall, we have reorganized the references to provide comprehensive background information and ensure their relevance to our paper.
(b): Abstract: We have restructured the language of the abstract to improve its conciseness and clarity. We have removed extraneous statements while retaining the core meaning and refined the language to enhance its connotation. For instance, we have abbreviated the background information in the first sentence of the abstract while preserving its meaning.
(c) Study design: We have reorganized some sections of our paper to enhance the study design's coherence and validity. Firstly, we have added markers for the corresponding data processing steps to Figure 1, the model mechanism diagram. Secondly, we have conducted a comparative analysis of the impact of reconstructing the length of 128-time sampling points and 32-time sampling points in the analysis section of the example and assessed the model's reliability based on MAE and MSE rise percentages. We have added Section 5.3.4 and Figures 11 and 12 to reflect these changes.
Additionally, we have reviewed the literature of similar work to compare the soundness of the research design. For instance, literature [1] proposes an LSGAN network for reconstructing power system data. The article first constructs the network and then uses an algorithm comparison to demonstrate the reliability of the model. The approach used in the algorithm analysis part is to compare different types of GAN implementations. Literature [2] uses gated neural units (GNN) to reconstruct the missing values in the logging data and finally also uses the comparison of models of the same type to compare the rationality of the proposed model. Literature [3], [4] is similar to this paper in terms of modeling principles, and both articles use the form of comparing different types of neural networks for the analysis of arithmetic cases. The structure of the article used in this paper is similar to the above pieces of literature, and the final arithmetic comparison is also in the form of comparing different types of neural networks, which makes the experimental structure of this paper reliable.
(d)Description method: The section on 1DCNN used in this paper is added in Chapter 4 of the article, the description of related contents is added, the working mechanism diagram about how 1DCNN is implemented is added, and the corresponding mechanism diagram is added in the attention mechanism implementation section. The main purpose of the modification is to make the description more detailed and comprehensive.
(e) Results Presentation:
- To achieve a balanced overall structure of the article, the data reconstruction structure chart and error analysis chart of 128 sampling time points were placed in the appendix. This prevents the issue of an asymmetric ratio of images to text caused by excessive images in the text.
- Each figure has been refined, with changes made to the aspect ratio, text size, and color scheme of the images. For the data reconstruction graphs, the model structure and actual values of this paper have been deliberately highlighted, while the results of other model reconstructions have been dashed. This enhances the clarity of the presentation of the results.
- Descriptions of the corresponding images have been added at the beginning of sections 5.3.1, 5.3.2, and 5.3.3, and the language of the results analysis section has been revised to improve the overall fluency and authenticity of the language. A comparative analysis of the reconstruction effect for 32 sampling time points was added at the end of 5.3.1 to enhance the analysis conclusion. 5.3.2 includes a comparative analysis of the reconstruction effect for 128 sampling time points to further improve the analysis. 5.3.3 reorganizes the logical structure and adds a summary of the corresponding images at the beginning of 5.3.3. Each paragraph is closely related to the corresponding data set, and a more detailed analysis of the error results is included in each paragraph.
- Subsection 5.3.4 focuses on the reconstruction results of 128 sampling points compared to the reconstruction results of 32 sampling points. It primarily analyzes the variation of two indexes, MSE and MAE, under different reconstruction lengths. Two comparison graphs, Figure 11 and Figure 12, are included correspondingly.
(f) Conclusion: The conclusion's logic has been reorganized, and the work done in this paper has been briefly summarized. The conclusions drawn from the work done in this paper are listed in points.
Point 2: It lacks innovation. The mentioned issue is a data recovery method for big data. It can be applied for all kind of data.
Response 2: Thank you very much for your comments. The work we have done mainly includes: 1. Adding the improved VMD algorithm to the data processing, so that the data can be processed before being sent to the deep learning model, which further improves the processing ability of the data model. 2. Introducing a triple attention mechanism of feature timing and multi-model combination based on Seq2Seq. In order to independently mine the correlation between the time series data output and each feature value, the feature attention mechanism is introduced to calculate the contribution rate of each feature quantity in real-time, and the feature weight is corrected; 3. At the same time, in order to mine the correlation between time-series information and the time-series data output and historical information, a time-series attention mechanism is introduced to independently extract historical key point information, improving the stability of long-term sequence reconstruction effects; 4. To make full use of the time structure characteristics of historical data, a multi-model combined attention mechanism is introduced. The multi-model combination attention mechanism can adaptively give reasonable weights to the output results of the two models, making the reconstruction result closer to the real value.
Our proposed model is built with reference to the characteristics of the power system data. For instance, the power system time series data contains many periodic laws, as it is closely related to human living habits and seasonal changes. Therefore, the VMD algorithm is used to extract the periodic law of the power system data center. Secondly, the power system data often includes other relevant characteristic values. For example, the photovoltaic power generation data often includes weather data and equipment parameters at the corresponding time in addition to power data. These data also have certain reference values for data reconstruction, but the importance is not the same. Therefore, the feature attention method is used in the article to give different weights to each feature value. Thirdly, the time series numbers of the power system contain many time correspondences. For example, when we reconstruct the load data, the data 24 hours ago has a strong reference value, which requires time series attention to identify its key points and give higher priority to it. Finally, there are relatively complete data on both sides of the missing data, which can be modeled separately on both sides and then the output results can be reasonably combined. All in all, the model-building process in this paper is based on the characteristics of the power system data.
[1] C. Wang, Y. Cao, S. Zhang, and T. Ling, “A Reconstruction Method for Missing Data in Power System Measurement Based on LSGAN,” Front. Energy Res., vol. 9, p. 651807, Mar. 2021, doi: 10.3389/fenrg.2021.651807.
[2] C. Jiang, D. Zhang, and S. Chen, “Handling missing data in well-log curves with a gated graph neural network,” GEOPHYSICS, vol. 88, no. 1, pp. D13–D30, Jan. 2023, doi: 10.1190/geo2022-0028.1.
[3] Y. Xiao, C. Zou, H. Chi, and R. Fang, “Boosted GRU model for short-term forecasting of wind power with feature-weighted principal component analysis,” Energy, vol. 267, p. 126503, Mar. 2023, doi: 10.1016/j.energy.2022.126503.
[4] M. F. Alsharekh, S. Habib, D. A. Dewi, W. Albattah, M. Islam, and S. Albahli, “Improving the Efficiency of Multistep Short-Term Electricity Load Forecasting via R-CNN with ML-LSTM,” Sensors, vol. 22, no. 18, p. 6913, Sep. 2022, doi: 10.3390/s22186913.

Reviewer 2 Report
A Power System Measurement Data Recovery Method Based on Improved VMD and Attention Mechanism Bi-directional CNN-GRU
The conributions and motivations should be clear and informative
In introduction the authors can come up with the existing survey works on the similar topic, probably summary table.
The research method is not clear, please clarify the research method involved.
Authors need to confirm that all acronyms are defined before being used
Based on the spectral properties of the input data being examined, how to choose the appropriate frequency interval threshold T1 and amplitude threshold T2
Due to the stability of electricity consumption habits, why the power system data possesses apparent repeatability.
Author Response
Response to Reviewer 2 Comments
Dear Editor and Reviewers,
We appreciate you for your precious time in reviewing our paper and providing valuable comments. It was your valuable and insightful comments that led to possible improvements in the current version. The authors have carefully considered the comments and tried our best to address every one of them. We hope the manuscript after careful revisions meet your high standards. The authors welcome further constructive comments if any. Below we provide the point-by-point responses. All modifications in the manuscript have been highlighted in red.
Statement of Revision
Point 1: The conributions and motivations should be clear and informative
Response 1: Thank you very much for your input. I made three revisions to your question. Firstly, the language of the abstract has been reorganized. The previous summary statement was too long and not concise enough. On the basis of retaining the core meaning, we deleted some sentences in the abstract and made a language polish on the abstract to make it more concise. For example, we abbreviated the background introduction of the first sentence of the abstract to make the language more concise without changing the expression. Secondly, we updated the references, deleted the parts that are weakly related to this article, and supplemented the parts that are related to this article in recent years. For example, documents 1, 2, and 4 were removed, and new documents were added. We also added some supporting references to the content of this paper, such as the literature of GRU and attention mechanism. In this way, we make the citations more contextual and relevant. Thirdly, we re-edited the conclusion and listed the conclusion in points to make it clearer.
Point 2: In introduction the authors can come up with the existing survey works on the similar topic, probably summary table.
Response 2: Thank you for your comments! I have added a summary table of some related work in the appendix of the article, which describes the technical means and advantages and disadvantages of the article, and give guidance at the end of the first chapter.
Point 3: The research method is not clear, please clarify the research method involved..
Response 3: Thank you for your comments! We have restructured parts of our article to make the research design more reasonable. Firstly, we added the identification of the corresponding data processing steps in Figure 1, which is the model mechanism diagram. This makes Figure 1 more closely integrated with the following data processing flow. Secondly, in the example analysis section, we also added a comparative analysis section of the reconstruction length effect of 128-time sampling points and 32-time sampling points and analyzed the reliability of the constructed model from the perspective of MAE and MSE increase percentage. Correspondingly, Section 5.3.4, Figure 11, and Figure 12 are added. In the fourth chapter of the article, the 1DCNN part used in this article is added, and the description of the relevant content is added. At the same time, the working mechanism diagram of how 1DCNN is implemented is added, and the corresponding mechanism diagram is added in the attention mechanism implementation part. The main purpose of the modification is to make the method described in this paper clearer. The pictures that appear are described in the article.
Point 4: Authors need to confirm that all acronyms are defined before being used
Response 4: Thanks for your comments, we reviewed the full text and made revisions to confirm that all acronyms are defined.
Point 5: Based on the spectral properties of the input data being examined, how to choose the appropriate frequency interval threshold T1 and amplitude threshold T2
Response 5: Thanks for your input. Since the spectrograms presented by different input data are quite different, it is difficult to quantify a definite formula for adaptation. Therefore, the selection of the frequency interval threshold T1 and the amplitude threshold T2 is somewhat subjective. The selection of frequency interval threshold T1 and amplitude threshold T2 in this paper is different for different data sets. For example, for load data sets, T2 selects 30% of the maximum amplitude. For T1, an extreme point interval of 30% of the maximum amplitude was chosen.
Point 6: Due to the stability of electricity consumption habits, why the power system data possesses apparent repeatability.
Response 6: Thank you for your input. Given the criticality of power systems to human life, it follows that the changing law of the power system is closely intertwined with the daily experiences of human beings. For instance, the change rule governing power system load data within a particular region remains relatively fixed on any given day, owing to the regularity of human work and rest patterns within that region. This results in a similarity between electricity load data observed on Monday and Tuesday. Moreover, this rule persists when the time scale is expanded to encompass an entire week. It is worth noting that similar patterns exist for other types of data, albeit with varying influencing factors.

Reviewer 3 Report
Here are my comments:
1. What's the meaning of both sides? before and after the missing data?
2. I am a little concerned about the portion of the testing data. 5% as testing data seems to be inadequate.
3. How did you select the model parameters? Did you implement any hyperparameter tuning to achieve the best performance?
4. What's the percentage of missing data?
5. I am a little unclear about the purpose of reconstructing the missing measurement data. Why this is critical for the power system operation?
Author Response
Response to Reviewer 3 Comments
Dear Editor and Reviewers,
We appreciate you for your precious time in reviewing our paper and providing valuable comments. It was your valuable and insightful comments that led to possible improvements in the current version. The authors have carefully considered the comments and tried our best to address every one of them. We hope the manuscript after careful revisions meet your high standards. The authors welcome further constructive comments if any. Below we provide the point-by-point responses. All modifications in the manuscript have been highlighted in red.
Statement of Revision
Point 1: 1. What's the meaning of both sides? before and after the missing data?
Response 1: Thank you. Typically, missing data within power systems are identified after the data has been generated. This results in a period of null values amidst a continuous stream of data. For instance, there may be an absence of electricity load data between the 5th and 6th hours within 24 hours. However, the data from hour 0 to hour 5 and from hour 6 to hour 24 remains intact. Thus, both sides refer to the segments of data on either side of the missing values within the power system. Temporally speaking, these are the data sets preceding and succeeding the period of missing values.
Point 2: I am a little concerned about the portion of the testing data. 5% as testing data seems to be inadequate.
Response 2: Thank you for your inquiry. In the calculation portion of my analysis, I utilized a total of three data sets. The load data set contained approximately 870 test samples, while the wind power generation and photovoltaic data sets contained roughly 2500 and 870 test samples respectively. This amounts to a total of approximately 4900 test samples. Although this may not seem like a large proportion, it is statistically significant in terms of quantity. Furthermore, during my training process, the loss values for both the test and validation sets were relatively similar, indicating that a 5% sample size should suffice for the selected data sets.
Point 3: How did you select the model parameters? Did you implement any hyperparameter tuning to achieve the best performance?
Response 3 :Thank you for your question. I considered hyperparameter optimization when building the model, but it was not pointed out in the article. Tuning my parameters with the help of the KerasTuner tool.
Point 4: What's the percentage of missing data?
Response 4:Thanks for your question, the proportions with missing data are 6.67% and 26.67%, respectively. Since the length of my refactored result is not the same
Point 5:I am a little unclear about the purpose of reconstructing the missing measurement data. Why this is critical for the power system operation?
Response 5:As the integration of intelligent equipment and renewable energy sources within power systems continues to increase, these systems are undergoing a transformation toward greater intelligence. The vast quantities of data generated by grid operations can provide valuable insights into the grid’s operational status, resilience, and potential future failures. However, missing data is an inevitable occurrence within grid data sets. The relative completeness of this data is crucial for effective big data processing. As such, the reconstruction of missing data within power systems plays a pivotal role in grid data analysis and represents the first step in this process.
Finally, we thank you again for your support of this article, and we look forward to your approval of this article and wish you well in your work!

Round 2
Reviewer 1 Report
The reviewers concerns have been addressed.
Reviewer 3 Report
The authors have addressed all my concerns, no further comments